# Monitoring Transepidermal Water Loss and Skin Wettedness Factor with Battery-Free NFC Sensor

**DOI:** 10.3390/s20195549

**Published:** 2020-09-28

**Authors:** Syed Muhammad Ali, Wan-Young Chung

**Affiliations:** Department of Electronic Engineering, Pukyong National University, Busan 48513, Korea; sm_ali@pukyong.ac.kr

**Keywords:** near field communication (NFC), transepidermal water loss (TEWL), skin wettedness factor (SWF), smart skincare sensor device tag, android application interface

## Abstract

The transepidermal water loss (TEWL) and the skin wettedness factor (SWF) are considered parts of a key perspective related to skincare. The former is used to determine the loss of water content from the stratum corneum (SC), while the latter is used to determine the human skin comfort level. Herein, we developed two novel approaches: (1) determination of the TEWL and the SWF based on a battery-free humidity sensor, and (2) the design of a battery-free smart skincare sensor device tag that can harvest energy from a near field communication (NFC)-enabled smartphone, making it a battery-free design approach. The designed skincare device tag has a diameter of 2.6 cm and could harvest energy (~3 V) from the NFC-enabled smartphone. A series of experimental tests involving the participation of eight and six subjects were conducted in vivo for the indoor and outdoor environments, respectively. During the experimental analysis, the skin moisture content level was measured at different times of the day using an android smartphone. The TEWL and SWF values were calculated based on these sensor readings. For the TEWL case: if the skin moisture is high, the TEWL is high, and if the skin moisture is low, the TEWL is low, ensuring that the skin moisture and the TEWL follow the same trend. Our smart skincare device is enclosed in a 3D flexible design print, and it is battery-free with an android application interface that is more convenient to carry outside than other commercially available battery-based devices.

## 1. Introduction

The human body’s skin can be divided into three depending layers: the hypodermis layer, the overlying dermis and the epidermis. The epidermis is the outermost layer of our skin, comprising the inner and outer parts. The outer part is called the stratum corneum (SC) [1] (Figure 1). SC initiates the skin moisture; therefore, it is required to determine the level of skin hydration and other parameters related to its health. Previously, multimodal sensors with soft, ultrathin and skin-like formats were developed for monitoring the hydration level of the skin [2]. Similarly, a wireless epidermal sensor based on passive inductive coupling was reported [3], where the analyzer was connected to a hand-wound copper primary coil, used to measure the hydration of the skin. Recently, an ultra-thin, tape free e-tattoo type sensor was designed, which could measure three features from the human skin, i.e., an electrocardiogram (ECG), the skin temperature and the skin hydration [4]. Similarly, a machine-learning-based noninvasive solution has been purported that uses galvanic skin response (GSR) to detect the hydration level in the human body [5]. The measurements of the transepidermal water loss (TEWL) and the skin wettedness factor (SWF) have been given significant importance in many research areas like dermatology, pharmacology, clinical analysis and cosmetic science. In addressing dry skin problems, one could obtain information about the TEWL rate from the skin, in order to make a choice in selecting relevant cosmetic products. However, individual assessments based on feeling are too subjective, which is why SWF is used to actually determine the human skin comfort level [6]. A portable tester giving information about the TEWL and the SWF could be conveniently used at home or outside to provide guidance in the selection of cosmetics. For instance, when showering and bathing, the proper kinds of soap or shower foam must be chosen according to the individual’s skin status, which could make the TEWL be low throughout the day. 

It is obvious that the TEWL cannot be measured directly [7], but it can be measured from the water vapor density at the skin surface (i.e., at the SC). Typical measurement method devices include open-chamber and condenser-chamber devices, which have been compared with each other according to their performances [7,8,9,10]. No standard reference has been given for the TEWL values, and thus a big difference can be found in the calibration of the TEWL measuring instruments, which depends solely on the device and the manufacturers [11]. The TEWL level also depends on the skin type. A lower TEWL is related to healthy skin, while a higher TEWL is related to damaged skin [12]. In this study, we took eight subjects inside and six subjects outside to monitor their skin’s TEWL. The TEWL is affected by many factors [13,14,15], including individual and environmental factors like age, environmental conditions, race of a person, sex, skin temperature and smoking. No direct method can be used to obtain the skin’s TEWL [1]; therefore, an efficient system which could calibrate the change of impedance in a moisture-measuring device with the loss of mass during the drying process was previously developed [1,16]. Similarly, in another reference, the TEWL was calculated with commercially available evaporimetry devices [17]. In our TEWL measurements, we adopted an almost similar approach, but the measurement was based on a battery-less techniques, i.e., involving no battery source in the device.

The SWF is actually used to determine the human skin comfort level [6]. The SWF depends on the relative humidity (RH) and temperature at the skin surface, and the ambient humidity (AH) and ambient temperature (AT) of the air. The increase in the SWF is responsible for introducing thermal discomfort, while it is independent of the body temperature [18]. Fukazawa et al. reported a linear relationship between the SWF and thermal discomfort [6]. Skin cooling and thermal sensations could drastically affect the SWF measurement, where the cooling rate threshold for identifying it as wet is based on the heat transfer rate from the skin [19]. The skin wettability must be determined based on physio-chemistry, considering that the physiochemical behavior of a human requires much attention [20]. Normally, as per the references [6,21,22], the SWF should be less than or equal to 0.3 in order to ensure a normal comfort level. However, Nishi et al. [23] suggested that the SWF should be expressed as the decimal fraction, 0.06 representing the minimum value, with 1 representing the maximum value for a fully wet skin. A number of research studies were previously conducted to analyze insights on the SWF that can be sensed in humans [24,25,26,27,28,29].

Near field communication (NFC), which is a technology launched in 2004 by three well-known companies (i.e., Nokia, Philips and Sony), constituted the name “NFC–Forum”. However, it was mostly adopted for payment systems. The NFC basically works on electromagnetic induction to establish communication between devices, which operate on an ISM band with a 13.56 MHz frequency [30], within a few centimeters (~4 cm). NFC communication consists of the reader and a tag. With the introduction of the NFC chip in the latest smartphones, NFC has gained serious interest in terms of its usage in Internet of Things (IoT) applications [31,32,33]. Energy harvesting through RFID or NFC is considered to be the best choice for embedded system applications. Several companies, like TI-, AMS-, NXP- and ST-Microelectronics, commercially introduced very low-power NFC-integrated chips (ICs) to ensure enhanced data writing, reading and storing capabilities. Utmost importance has recently been given to energy harvesting and data communication from NFC, with a focus on the optimal ways to obtain some future directions [34,35,36]. This study designs a smart skincare device with a novel design approach that could harvest energy from an NFC-enabled smartphone and elaborates on the applications related to skincare.

## 2. Materials and Methods

### 2.1. TEWL Calculation

The TEWL is generally computed in gm/cm^2^·s, and it refers to the weight (gm) of moisture per unit time (s) per unit area (cm^2^). It is defined as the amount of moisture that evaporates from 1 cm^2^ of the skin surface per unit second. The TEWL has been given prime importance in dermatology, but it cannot be determined by the naked eye [37]. According to reference [38], the TEWL is described by Fick’s second law, as follows:(1)$$T\;=\;D\;\frac{d}{dx}C$$
where *D* is the diffusion of water at the SC, expressed in cm^2^/s; *C* is the amount of water concentration in the SC; and *x* represents the skin membrane thickness (15 × 10^−4^ cm) from the edge of the SC. The value of *D* is suggested to be 10^−9^ cm^2^/s [38], and the value of *C* can be calculated as [39]:(2)$$C\;=\;\frac{4.39\;\times \;10^{-2}\;\times \;RH}{100}\;$$
where *RH* is the value of the skin moisture taken from the humidity sensor embedded on the skincare device tag.

### 2.2. SWF Calculation

The sensor used in our experiment could also detect the skin temperature; therefore, the skin temperature and moisture are the key parameters for determining the SWF, using the following formula [40]:(3)$$\mathrm{SWF}\;=\;\frac{\rho_{s}\;-\;\rho_{a}}{\rho_{s-sat}\;-\;\rho_{a}}\;$$
(4)$$\mathrm{SWF}\;=\frac{Rh_{skin.}\rho_{skin-sat}\;-\;Rh_{a.}\rho_{a-sat}}{\rho_{skin-sat}\;-\;Rh_{a.}\rho_{a-sat}}$$
where $Rh_{skin}$ represents the RH at the skin surface, which is actually the skin moisture; $\rho_{s}$ is the partial vapor pressure at the skin surface; $\rho_{a}$ is the partial vapor pressure in the ambient air; $\rho_{skin-sat}$ is the saturated vapor pressure at the skin surface; $\rho_{a-sat}$ is the saturated vapor pressure of the air at an AT; and $Rh_{a}$ represents the RH of the air, which is actually the AH. The saturated vapor pressure at the skin surface can be determined as [40]:(5)$$\rho_{skin-sat}\;=\;0.13332\;\times \;10^{8.07\;-\;\frac{1730.63}{T-39.72}}$$

Similarly, the saturated vapor pressure of the air can also be found from [40]:(6)$$\rho_{a-sat}\;=\;0.13332\;\times \;10^{8.07\;-\;\frac{1730.63}{T-39.72}}$$

### 2.3. System Design

The overall system design structure of the skincare sensor device tag included an NFC-enabled android smartphone, an embedded NFC coil on the skincare device tag to harvest sufficient energy from the smartphone, an NFC transponder chip for data communication with the smartphone, an ultra-low power microcontroller (MCU), schottky diodes for the rectification process and a skin moisture sensor (Figure 2a). In Figure 2b, the main components embedded on the bottom layer of the FR4 substrate are depicted, making the device a smart skincare sensing device tag. The MCU (MSP430G2553, Texas Instruments Inc., Dallas, TX, USA) could consume different amounts of power (1.8–2.2 V at 230 µA) while working on different clock speeds (typically around 1 MHz). The NFC transponder chip (RF430CL330H, Texas Instruments Inc., Dallas, TX, USA) is registered with the ISO14443B standard and uses the NFC data exchange format (NDEF) message with a data rate of 848 kbps, while writing data to the NFC smartphone. An I^2^C or SPI message protocol is required by this NFC chip to communicate with the MCU and a sensor while writing data to the NFC smartphone. The chip consumed 250 µA while operating on a 400 KHz frequency. Moreover, the skin moisture sensor (BME 280, Bosch Sensortec, Germany), which calculated the TEWL and the SWF, required only 1.8 µA at 1 Hz to operate efficiently on the humidity and temperature modes.

The fabricated smart skincare device tag measured 2.6 cm in diameter (Figure 3a) and was enclosed on a flexible 3D-printed object (Figure 3b), enabling it to operate fully and conveniently in a passive mode. This lightweight 3D flexible structure made the device more convenient to take outside, especially for those people who are often quite anxious when it comes to the usage of any skincare device. The operating distance for energy harvesting from the NFC-enabled smartphone was approximately 1 cm (Figure 3c), making the skincare device tag work efficiently on battery-less techniques. The energy harvesting efficiency depends on a number of factors, including the antenna dimensions. In our case, the six-turn coil antenna embedded on the skincare device tag had the following dimensions: thickness of each turn coil: 0.02 cm; total inner radius of the circular coil antenna: 1 cm; and total outer radius of the circular coil antenna: 1.25 cm. The maximum harvested energy was measured with an oscilloscope (DS07054A, Tektronix Inc., Beaverton, OR, USA). The peak–peak (Pk–Pk) harvested voltage was 3.06 V when the smartphone was brought very close to the skincare sensor tag device (~1 cm); however, it gradually decreased as we started moving the smartphone away from the skincare device tag (>1 cm) (Figure 4).

For the skincare device to operate effectively and efficiently, the six-turn coil antenna was first designed and tested on a well-known magnetic simulator (Ansoft Maxwell, ANSYS Inc., Canonsburg, PA, USA) to study the electromagnetic simulations. The inductance (3.5887 µH) of the fabricated antenna coil on the device was then measured with a network analyzer (E5062A, Agilent Technologies Inc., Santa Clara, CA, USA) (Figure 5) to determine the tuning capacitor (*C* = 1/4π^2^f^2^L = 38.38 × 10^−12^ ≈ 38 ρF) that should be soldered on the fabricated PCB for proper tuning at the desired frequency (13.56 MHz). More importantly, before finding the coil inductance, all the components were properly soldered on the FR4 substrate to ensure that all the parasitic capacitances and inductances, which might affect the result of finding the inductance of the skincare device coil, were also added.

### 2.4. Android Application Design

A smart skincare app was designed on the android platform. This app could display the percentage values of the skin moisture taken from the sensor in a circular bar, as well as the skin moisture level (Figure 6a). After taking the skin moisture sensor readings, the values were processed to calculate the corresponding TEWL and SWF values. In android, waterfall graphs with the trendline were later displayed to show the history of the maximum and minimum values of TEWL and SWF in the indoor and outdoor environments, registered at different times of the day. The highest TEWL value noted at 10 AM in indoor environment among the eight subjects was 0.0215 µgm/cm^2^·s, while the lowest one was 0.0162 µgm/cm^2^·s (Figure 6b). Similarly, at 2 PM, 4 PM and 6 PM, the highest (lowest) TEWL readings recorded in the android smartphone were 0.0193 µgm/cm^2^·s (0.0148 µgm/cm^2^·s), 0.0167 µgm/cm^2^·s (0.0139 µgm/cm^2^·s) and 0.0163 µgm/cm^2^·s (0.0109 µgm/cm^2^·s), respectively, in the indoor environment (Figure 6b). The same process was repeated for the outdoor environment, illustrating the highest (lowest) TEWL readings at 10 AM, 1 PM, 3 PM and 5 PM to be 0.0163 µgm/cm^2^·s (0.0117 µgm/cm^2^·s), 0.0170 µgm/cm^2^·s (0.0099 µgm/cm^2^·s), 0.0174 µgm/cm^2^·s (0.0119 µgm/cm^2^·s) and 0.0164 µgm/cm^2^·s (0.0119 µgm/cm^2^·s), respectively (Figure 6c). The TEWL trendline followed a decreasing fashion in the indoor environment. In contrast, an increasing trend, especially during the afternoon times, was observed in the outdoor environment because most of the subjects felt sweaty during that period. The same process was repeated for the SWF calculations. The highest (lowest) SWF values recorded at 10 AM, 2 PM, 4 PM and 6 PM for the indoor environment were 0.41 (0.16), 0.47 (0.14), 0.37 (0.14) and 0.38 (0.01), respectively (Figure 6d). Meanwhile, the highest (lowest) SWF values recorded at 10 AM, 1 PM, 3 PM and 5 PM for the outdoor environment were 0.51 (0.35), 0.53 (0.21), 0.52 (0.29) and 0.48 (0.32), respectively (Figure 6e). Figure 6c,e show that the TEWL and the SWF in the outside environment followed an almost similar trendline; that is, the higher the TEWL, the higher the SWF, and the lower the TEWL, the lower the SWF.

## 3. Results

### 3.1. TEWL in the Indoor and Outdoor Environments 

After soldering the components on the PCB and enclosing the device on a flexible 3D object, experiments were conducted in both indoor and outdoor environments by using the smart skincare sensor device tag. The skincare device tag harvested sufficient energy from a well-enabled NFC smartphone. The skin moisture readings were taken through the sensor tag. These readings were then plugged into Equation (2) to calculate the water concentration at the SC. After finding C, the values were then inserted in Equation (1) to compute the TEWL levels. For the indoor experimental analysis, the TEWL readings on each subject were taken at 10 AM, 2 PM, 4 PM and 6 PM (Figure 7). Similarly, the TEWL readings for the outdoor experiment were taken at 10 AM, 1 PM, 3 PM and 5 PM (Figure 8). Some of the subjects felt sweatier outdoors between 2 PM and 4 PM; therefore, the increase in skin moisture was observed to be significant, and caused the TEWL to also increase during that time period (i.e., 3 PM to 5 PM).

The sensor readings from the humidity sensor and the calculated TEWL values were then compared, and it was observed that the histograms in Figure 9 and Figure 10 depicted how the skin moisture and TEWL values followed the same trend (i.e., the higher the skin moisture, the higher the TEWL rate, and the lower the skin moisture, the lower the TEWL rate).

### 3.2. SWF Measurements in the Indoor and Outdoor Environments

The sensor integrated in our skincare device tag could also measure the skin temperature; thus, the skin temperature values were inserted in Equation (5) to determine the saturated vapor pressure at the skin surface. The AT and AH readings were taken using a commercial meter (HTC-1 Clock/Humidity/Temperature, MCR Inc., Yueqing, Wenzhou, China). Equation (6) was then used to determine the saturated vapor pressure of the air. Finally, the values of the AH, RH at the skin (skin moisture), saturated vapor pressure at the skin surface and saturated vapor pressure in the air were all inserted into Equation (4) to calculate the SWF. The SWF values should be less than or equal to 0.3 to ensure a normal comfort level. The SWF was calculated depending on the experimental results in the indoor and outdoor environments. The results in Figure 11 and Figure 12 indicate that the subjects had different responses, with the range of the SWF being between 0.1 and 0.5.

## 4. Discussion

In this study, we developed a novel smart skincare sensor device tag that could monitor TEWL and SWF values, based on a battery-free humidity sensor capable of harvesting energy from an NFC-based smartphone. The TEWL and the SWF provide very useful, beneficial information for skincare in the cosmetics field and can be calculated by taking skin moisture and skin temperature readings from a battery-free skincare sensor device tag.

Our skincare device tag was compact, with small dimensions, (i.e., a diameter of only 2.6 cm), and could harvest ~3 V from the NFC-enabled smartphone, where the distance between the smartphone and the skincare device tag was approximately 1 cm. A series of in vivo experiments were conducted in both indoor (eight subjects) and outside (six subjects) environments to analyze the data. There was a big contrast in the experimental results of the indoor and outdoor environments. This meant that human skin was more vulnerable to an outside environment, and body sweat also affected skin moisture. Our skincare device tag was enclosed in a flexible 3D-printed object, had a high monitoring resolution and could be easily transported by subjects. This means that our skincare device tag is a very comfortable choice for customers when compared to other commercially available devices.

Moreover, an android application was developed to save skin moisture and skin temperature sensing values in an android smartphone. The app could comprehensively and easily provide the history of the calculated TEWL and SWF values for the user.

In the future, this smart skincare sensor device tag could be fabricated on a flexible design PCB, which could be more convenient for use, and, similarly, a considerable amount of effort may also be required in order to make the dimensions of the proposed flexible PCB be smaller and more compact.

## Figures and Tables

**Figure 1 sensors-20-05549-f001:**
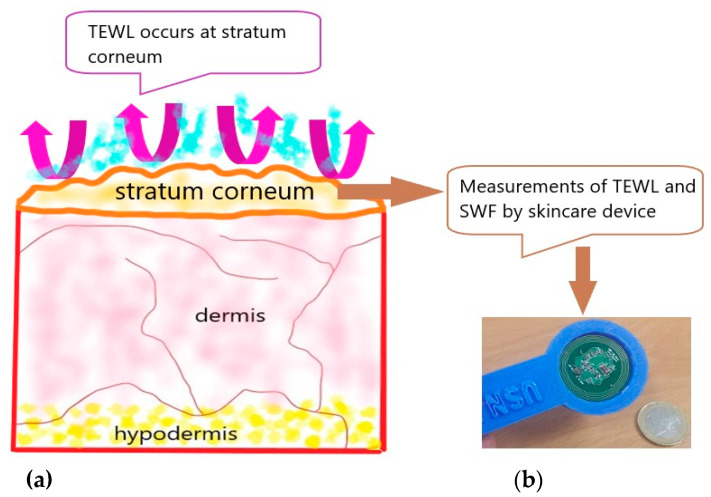
(**a**) Typical skin layer structure; (**b**) skincare sensor device tag.

**Figure 2 sensors-20-05549-f002:**
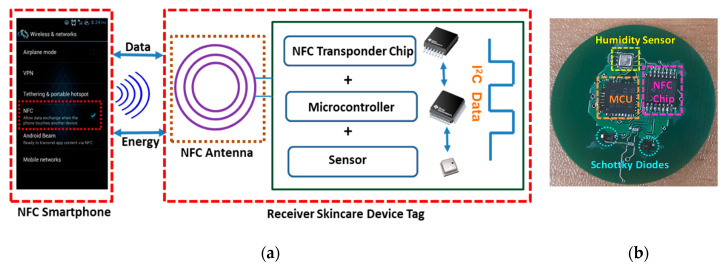
(**a**) Overall structure of the battery-free smart skincare sensor device tag system; (**b**) main components of the skincare device tag.

**Figure 3 sensors-20-05549-f003:**
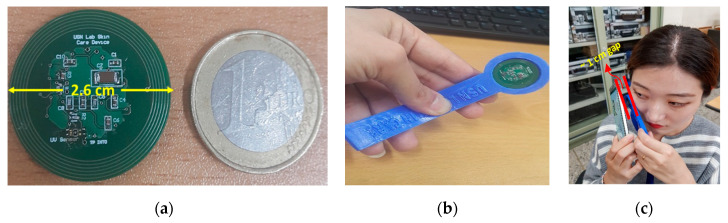
(**a**) Dimensions of the smart skincare sensor device tag; (**b**) skincare sensor device tag enclosed within the flexible 3D-printed object; (**c**) operating distance of the skincare sensor device tag from the NFC smartphone.

**Figure 4 sensors-20-05549-f004:**
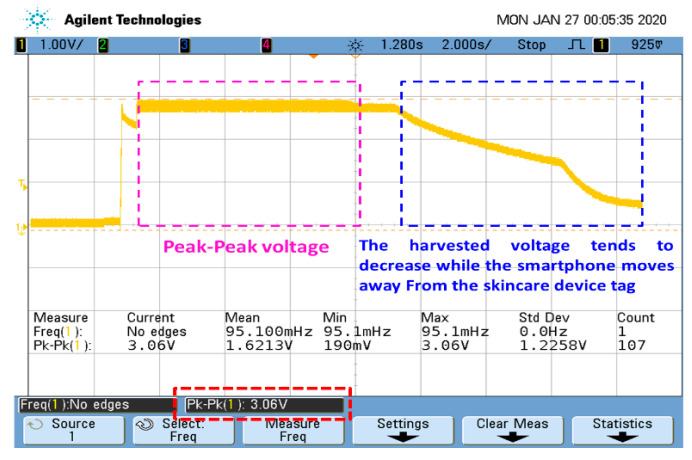
Maximum energy harvested from the NFC-enabled smartphone.

**Figure 5 sensors-20-05549-f005:**
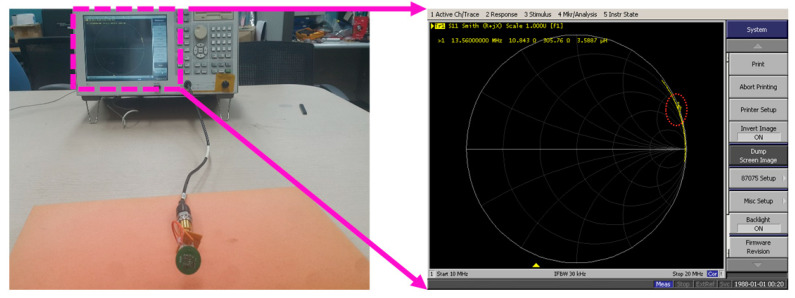
Inductance of the embedded coil in the smart skincare device tag.

**Figure 6 sensors-20-05549-f006:**
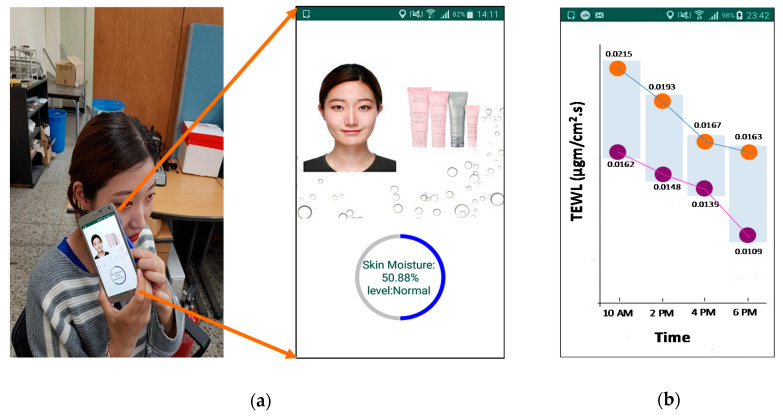
Android app used in combination with the smart skincare sensor device tag: (**a**) measurement of the skin moisture through a battery-free NFC approach; (**b**) waterfall graph with a trendline illustrating the TEWL result on the inside environment of eight subjects; (**c**) waterfall graph with a trendline illustrating the TEWL result on the outside environment of six subjects; (**d**) waterfall graph with a trendline illustrating the SWF result on the inside environment of eight subjects; and (**e**) waterfall graph with a trendline illustrating the SWF result on the outside environment of six subjects.

**Figure 7 sensors-20-05549-f007:**
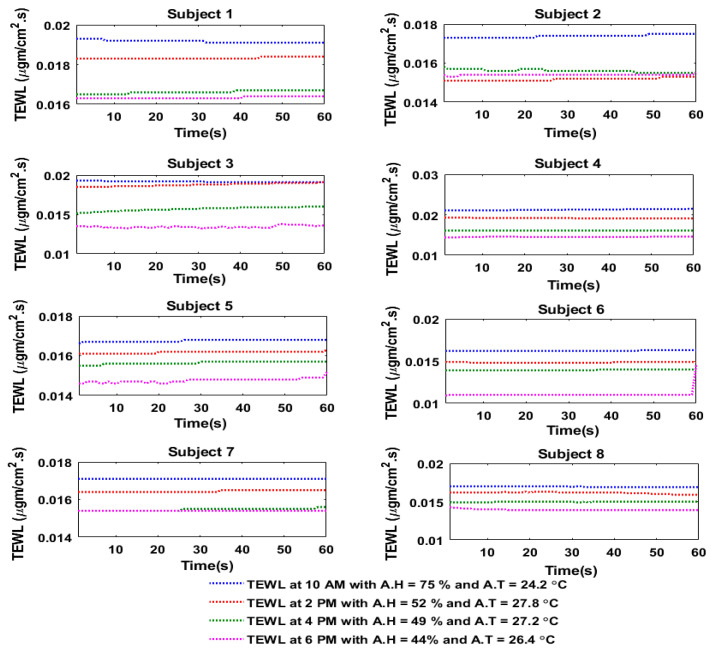
TEWL of the eight subjects at different times of the day during the indoor experiment.

**Figure 8 sensors-20-05549-f008:**
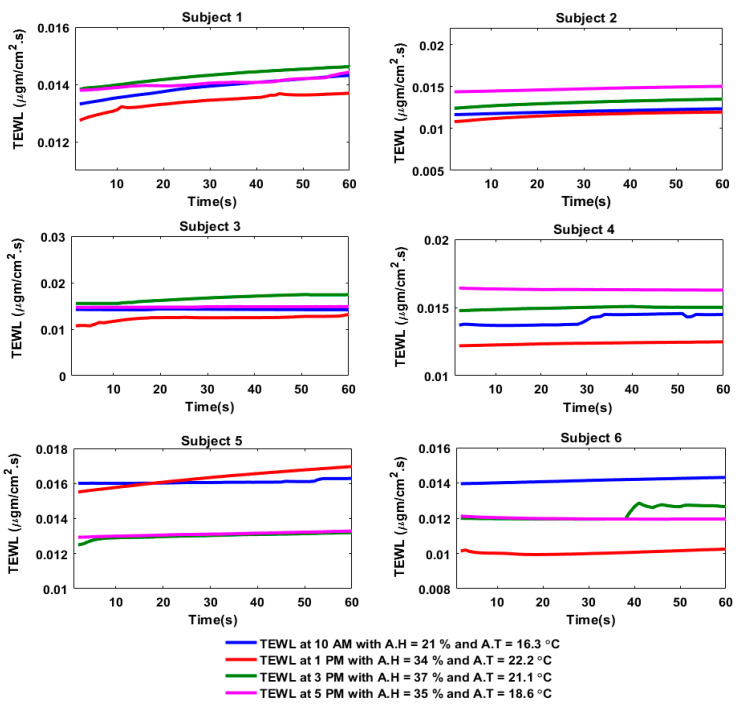
TEWL of the six subjects at different times of the day during the outdoor experiment.

**Figure 9 sensors-20-05549-f009:**
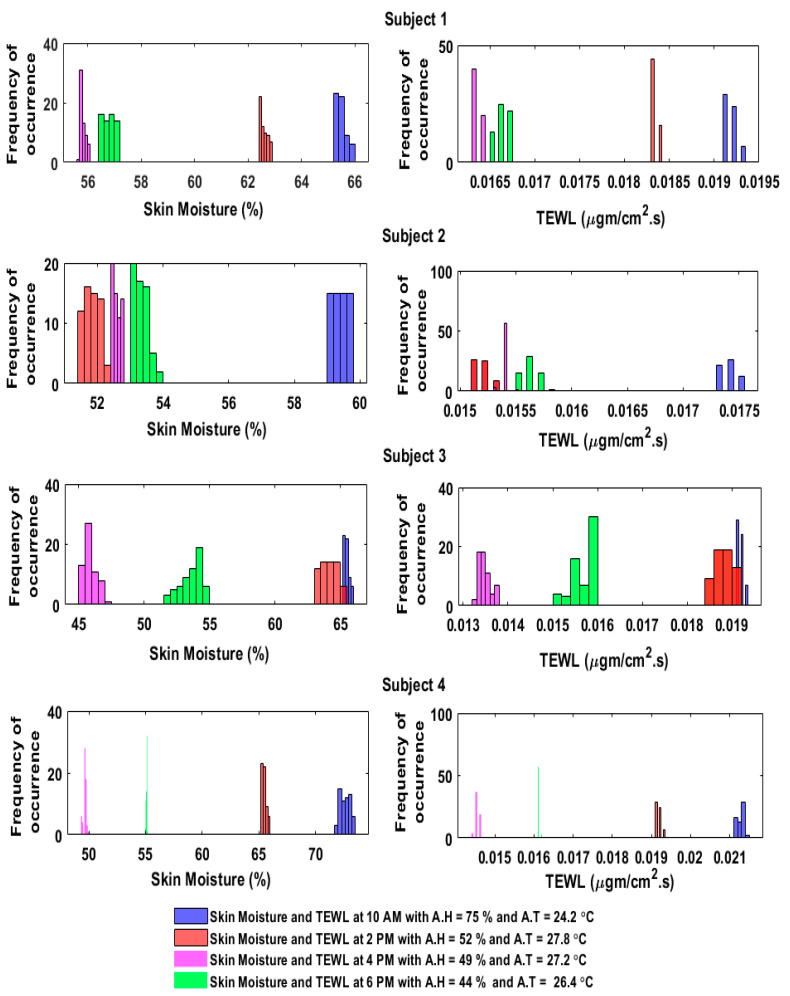
Histogram relationship between the skin moisture and the TEWL in the indoor environment.

**Figure 10 sensors-20-05549-f010:**
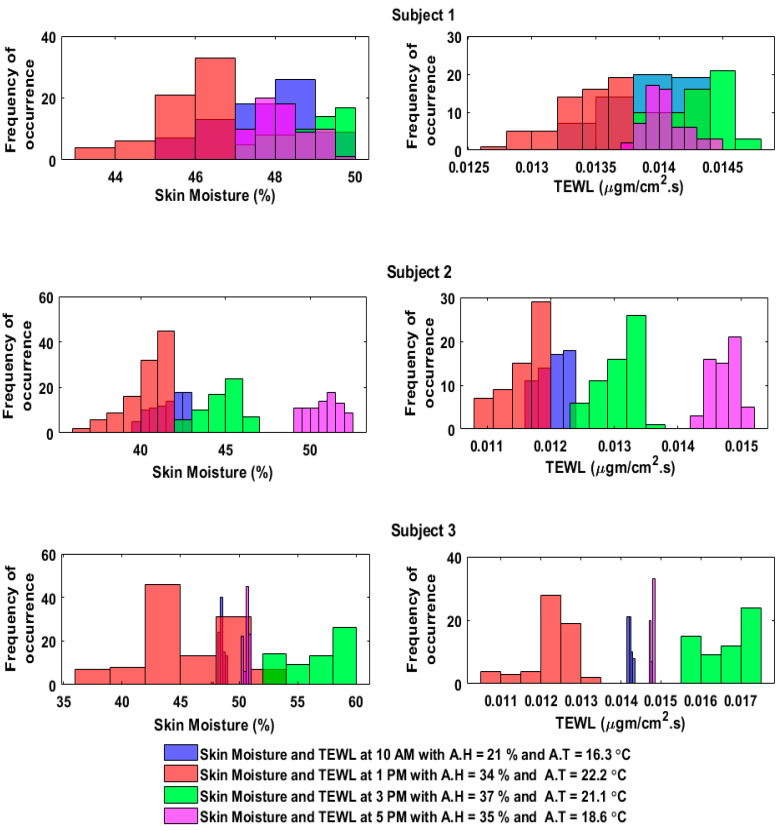
Histogram relationship between the skin moisture and the TEWL in the outdoor environment.

**Figure 11 sensors-20-05549-f011:**
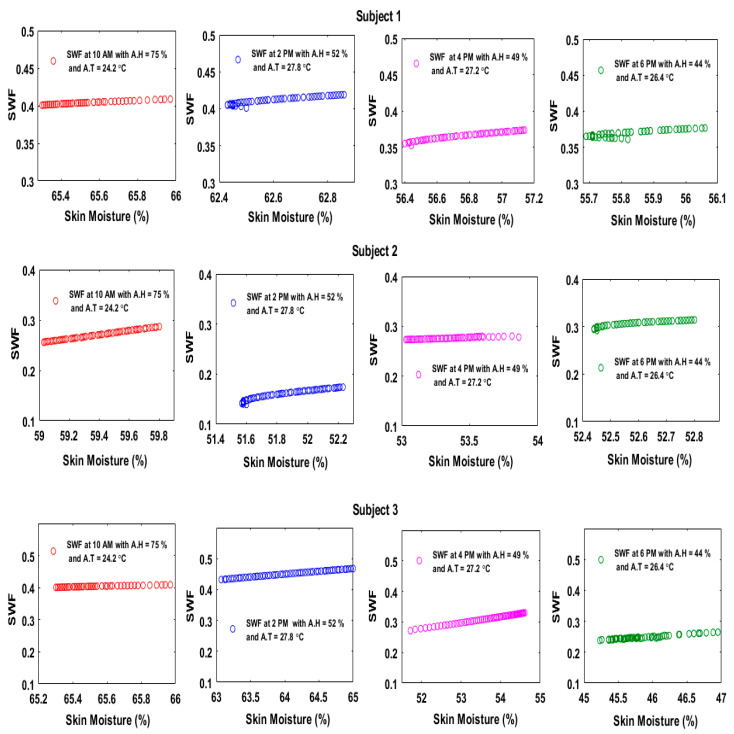
SWF of three subjects at different times of the day during the indoor experiment.

**Figure 12 sensors-20-05549-f012:**
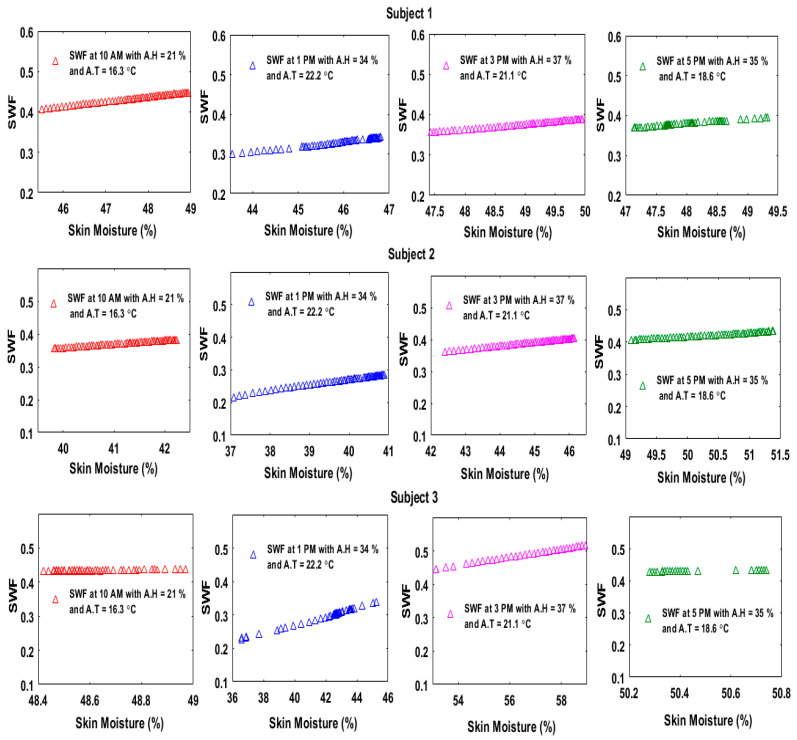
SWF of three subjects at different times of the day during the outdoor experiment.

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
