# Peer review of "Monitoring Transepidermal Water Loss and Skin Wettedness Factor with Battery-Free NFC Sensor"

_sensors, 2020, doi:10.3390/s20195549_

Round 1

Reviewer 1 Report

The authors developed a novel smart skincare sensor device tag that could monitor the TEWL and SWF values with a battery-free humidity sensor, which can harvest energy from NFC. The TEWL and the SWF help skincare in the cosmetics field. This work may interest readers from the field of wearable biosensors. It is publishable after answering the following questions.

  1. Which moisture sensor was used in this work? I don't see the information in the text. 
  2. For data collection, on which part of the body the sensor was attached? It is assumed that the value varies with location. How to precisely control the experiments. 
  3. What is the main advantage of the sensor compared to the commercial ones? It will be interesting to see a flexible demo. Can it be demonstrated? 
  4. How about stability? Besides, can it work well in the water environment (100% humidity)
  5. Figure 6 needs to be improved. It seems a screenshot. 

Author Response

The authors developed a novel smart skincare sensor device tag that could monitor the TEWL and SWF values with a battery-free humidity sensor, which can harvest energy from NFC. The TEWL and the SWF help skincare in the cosmetics field. This work may interest readers from the field of wearable biosensors. It is publishable after answering the following questions.

(Our response):

We are happy to see that the reviewer recommended our novel work in the statement(s) above, and suggested it to make it publishable.

1.Which moisture sensor was used in this work? I don't see the information in the text. 

(Our response) :

The moisture sensor used in our work is a humidity sensor i.e., BME 280(Bosch Sensortec, Germany), which is specifically designed for mobile application wearables, where the low power consumption, and size to be smallest is of paramount importance.

Moreover, this information has been inserted in the text of manuscript at page 4, line no. 139.

2. For data collection, on which part of the body the sensor was attached? It is assumed that the value varies with location. How to precisely control the experiments. 

(Our response):

The sensor in the skincare device tag was touched on the face of the skin of the subject to get the sensor readings, as taking the moisture data from the face is more appropriate in cosmetic science.

We didn’t assume anything related to measurements in the experimental data. The real original measurement data were taken during complete analysis. Surely, the values vary with change in location. At the equator and subtropical areas, particularly in the African countries, where the ambient humidity is high, the skin moisture values would be high as compared to the places where the ambient humidity is low.

It should be noted that the ambient humidity and ambient temperature affects the skin moisture. In the indoor environment, where the ambient humidity and ambient temperature is well controlled, the trend of data is different from the outdoor. However, the sensor provides the exact true skin moisture values, no matter it is indoor or outside because the sensor is enclosed in a closed chamber.

3. What is the main advantage of the sensor compared to the commercial ones? It will be interesting to see a flexible demo. Can it be demonstrated?   

(Our response):

The main advantage of the sensor (used in our skincare sensor device tag) as compared to others is that the sensor is enclosed in a closed chamber, and only a vent hole is available from where we can measure the skin moisture by touching it on the skin. Inside the vent hole, there is a capacitive electrode that measures the skin moisture based on a change in capacitance values. Moreover, this chamber protects from external environmental effects like ambient humidity etc. This means the measurements taken from the sensor is robust. Thus, it enables us to give a precise and accurate reading of skin moisture only through the vent hole. The sensor module is housed in an extremely compact metal lid LGA package with a footprint of only 2.5×2.5 mm2 and a height of 0.93 mm. 

4. How about stability? Besides, can it work well in the water environment (100% humidity)

(Our Response):

The sensor is having the absolute accuracy tolerance of ±3% while measuring the skin moisture, and the response time to complete 63% of the step is just 1 second, which shows the stability of the sensor is extremely high.

Surely, it can work well when the ambient humidity is 100% because only vent the hole is open which needs to be touched with the skin, while the rest of the sensor is enclosed in a robust closed chamber.

5. Figure 6 needs to be improved. It seems a screenshot. 

(Our response)

Fig. 6 is very important because it depicts the real time true data values taken by the android smartphone app. The development of the android app was done in android studio software using Java coding. The history of all the data, i.e. TEWL and SWF is being saved in an Android smartphone and displayed in the form of waterfall graphs. Since we are taking sensor data only through android smartphone, therefore it is necessary to add these screenshots in order to properly elucidate the experimental setup.

Reviewer 2 Report

1/The author did not provide the detail structure of the sensor. it is very important for application,for example, is it close to attach to the skin when use or nearly to? what is the duration to test each time? the measure process maybe disturbs the TEWL or SWF test.
2/ as a kind of sensor, the properties of repeatability stability did not presented, especially the errors. though some subjects data provided, but did not compared with other method, so the accuracy did not validated.
3/the manuscript shown little of the innovation in measuring skin TEWL and SWF if only address battery free techniques. Meanwhile, it is inconvenience for user compared with BT technique.

Author Response

1. The author did not provide the detail structure of the sensor. it is very important for application, for example, is it close to attach to the skin when use or nearly to? what is the duration to test each time? the measure process maybe disturbs the TEWL or SWF test. 

 (Our Response)

It is necessary to touch the sensor with the skin to take the real original sensor readings.

The duration to test each time is 60 seconds as shown in Figs. 7 and 8.

The sensor was programmed through embedded coding, thus it should provide data during each 1 second interval. After taking the skin moisture values during each second, the TEWL and SWF were calculated and processed immediately in the embedded systems; therefore, testing won’t disturb the TEWL or SWF values.

Moreover, the complete detailed structure has been explained in the reviewer 1 part at point 3.

2. As a kind of sensor, the properties of repeatability stability did not presented, especially the errors. though some subject data provided, but did not compared with other method, so the accuracy did not validated.

(Our Response)

The sensor used in our skincare device tag is having an accuracy of 97%, which is the most robust accuracy of any humidity sensor in the reference literature so far.

In order to measure the subject data, the analytical and experimental methods were used which is a completely novel approach. Analytical (mathematical formulas) are taken from the valid references which in our opinion doesn’t need to be compared with other methods.

3. The manuscript shown little of the innovation in measuring skin TEWL and SWF if only address battery free techniques. Meanwhile, it is inconvenience for user compared with BT technique.

(Our Response)

The main focus of this manuscript is to design a battery-free skincare device tag, and address its implication in the skincare domain. This is the first novel approach; the device is having the smallest compact dimension, i.e. 2.6 cm and is capable of harvesting its energy from the NFC smartphone. This provides a very convenient, smarter and best approach in the field of cosmetic dermatology to measure the TEWL and SWF values using a skin moisture sensor.

The novel application areas related to skincare, which we targeted in this manuscript, are TEWL and SWF; by working on the concrete background analytical analysis and implementing it on an experimental basis, which is in fact a big innovation.

We didn’t understand BT part in the question completely. Maybe you are asking about the Bluetooth module and using this BT module instead of the NFC chip. Please note that the device having Bluetooth can never ever be implemented using this battery-less technique because the Bluetooth needs a massive amount of power (approx. 3.3 V) to operate. Interestingly, the Bluetooth chip could just be used for the transfer of the data while the NFC chip provides dual purpose, that is, first energy scavenging (It uses 13.5 MHz frequency and work on electromagnetic induction phenomena to transfer the energy); secondly use NDEF message protocol for transfer of the data between the two devices. Moreover, the Bluetooth chip antenna inside the smartphone is not even capable of providing energy scavenging process. In our skincare domain in dermatology, the NFC based method was proved to be more appropriate and accurate because of the utilization of the NFC chip in a smartphone, which could provide the benefits in many folds as compared to BT.

Reviewer 3 Report

The authors present two new approaches developed with the aim of determining transepidermal water loss and skin wettedness factor using battery-free moisture sensors. They represent the design of a battery-free device for smart skin care that gets its energy from a smartphone with a near field communication option. The device elaborates on the applications related to skincare.

The paper clearly presents the problem to be solved and the methodology used to solve the problem. The results are generally interpreted in an appropriate way, except that a description of the collaboration with experts in the field of application and their role in the research is missing.

Reference 1 is not the most appropriate reference for a medical description of the skin and its characteristics because it is an article from the field of technical sciences (line 30). It is necessary to cite the original reference from the field of medical sciences.

As the article is from the field of technical sciences, it would be more appropriate for the introduction to start with the motivation for research in that field of science, rather than medical descriptions of the skin (this is the field of application). The paper lacks information on the role of experts in the field of application in the phase of setting hypotheses and in the phase of evaluation of results. That context needs to be added to the paper.

Line 35: Reference [4] from 2018 is cited as a reference to recent research, however, as this is a very active area of ​​research, it is necessary for the authors to cite at least one more recent reference.

In line 160 I found inappropriate and awkward the statement for a paper in a scientific journal: please change „for females“ to „for those“ of “for people“

Please discuss what are the shortcomings of your approach and what is the plan for the future work.

The technical quality of the work is good.

Author Response

1. The authors present two new approaches developed with the aim of determining transepidermal water loss and skin wettedness factor using battery-free moisture sensors. They represent the design of a battery-free device for smart skin care that gets its energy from a smartphone with a near field communication option. The device elaborates on the applications related to skincare.

The paper clearly presents the problem to be solved and the methodology used to solve the problem. The results are generally interpreted in an appropriate way, except that a description of the collaboration with experts in the field of application and their role in the research is missing.

(Our Response)

We are pleased that the reviewer recommended our manuscript work and appreciated the underlying methodology and the problem solving approach.

2. Reference 1 is not the most appropriate reference for a medical description of the skin and its characteristics because it is an article from the field of technical sciences (line 30). It is necessary to cite the original reference from the field of medical sciences.

(Our Response)

As per the reviewer’s suggestion, we have made the necessary changes and cited the original medical reference.

  1. Sahle, F.F.; Gebre-Mariam, T.; Dobner, B.; Wohlrab, J.; Neubert, R.H.H. Skin diseases associated with the depletion of stratum corneum lipids and stratum corneum lipid substitution therapy. Skin Pharmacol. Physiol. 2015, 28, 42–55.

3. As the article is from the field of technical sciences, it would be more appropriate for the introduction to start with the motivation for research in that field of science, rather than medical descriptions of the skin (this is the field of application). The paper lacks information on the role of experts in the field of application in the phase of setting hypotheses and in the phase of evaluation of results. That context needs to be added to the paper.

(Our Response)

Since this paper is written for a particular medical issue, therefore, in our opinion, we think it is more appropriate to start like this. The field of application depicts the concrete analytical method described from the reliable references, as illustrated in the manuscript, which in our opinion, doesn’t need any further elaboration or the role of experts to signify the importance of the methods and hypotheses.

4.Line 35: Reference [4] from 2018 is cited as a reference to recent research, however, as this is a very active area of ​​research, it is necessary for the authors to cite at least one more recent reference.

(Our Response)

As per the reviewer’s suggestion, one more reference from the year 2020 has been included and cited in the text. The information has been included in the manuscript at page 1, line no. 37.

    1. Rizwan, A. et al. Non-Invasive Hydration Level Estimation in Human Body Using Galvanic Skin Response. IEEE Sensors J. 2020, 20(9), 4891–4900.

5.In line 160 I found inappropriate and awkward the statement for a paper in a scientific journal: please change „for females“ to „for those“ of “for people“

(Our Response)

As per the reviewer’s suggestion, the required modification has been done as highlighted in the text of manuscript.

6.Please discuss what are the shortcomings of your approach and what is the plan for the future work.

(Our Response)

The shortcomings of our skincare device tag and adopted approach include: we have to use and touch the sensor on the skin for about 1 second during each time interval. We can’t touch the sensor on the skin for a prolonged period of time because more and more skin moisture gets accumulated inside the chamber, and measurement readings would slightly change, which might disturb the accuracy of the results.

The future work plan is to design this skincare device tag on the flexible design PCB, and similarly make the dimensions of this PCB more smaller and more compact.

Moreover, the information related to the future work has been included in the text of manuscript at page 14, line no. 349.

7.The technical quality of the work is good.

(Our Response)

We are very thankful for appreciating the technical quality of our work.

Round 2

Reviewer 2 Report

1、This manuscript is not good written as the author claimed "The main focus of this manuscript is to design a battery-free skincare device tag, and address its implication in the skincare domain. .“ its novelty is limited.

2、What are the challenges with NFC technique and how to reach the goals in this work? It is important to introduce in section 1.

3、In addition, the tests design of selection of subjects are not very reasonable, because lack of their age, occupation, sex etc. so that the data seams that it doesn‘t have more practical significance for skincare.

4、The manuscript looks more like an experiment report that analyzing and discussing related with NFC sensor should be strengthened.

Author Response

Dear Editor and the Reviewer,

                        Please have a look on the attachment, answering all the comments.

Thanks,

Wan-Young.
